# Parameterising the Impact of Roughness Evolution on Wind Turbine Performance

Jack Kelly * , Richard Willden and Christopher Vogel

Department of Engineering Science, University of Oxford, Parks Road, Oxford OX1 3PJ, UK;
richard.willden@eng.ox.ac.uk (R.W.); christopher.vogel@eng.ox.ac.uk (C.V.)
* Correspondence: jack.kelly@eng.ox.ac.uk

**Abstract:** This paper presents a study investigating the effects of surface roughness on airfoil performance and its consequences for wind turbine energy yield. This study examined 51 sets of experimental data across 16 airfoils to identify trends in roughened airfoil performance. The trends are used to formulate a novel 'roughness evolution parameter' that can be applied to airfoils with no roughened data available to predict the impact of roughness on performance. Blade element momentum theory is used to model the performance of the DTU 10 MW reference wind turbine, with uniformly roughened blades emulated using the roughness evolution parameter. An annual energy production loss between 0.6–9.6% is found for the DTU 10 MW turbine when considering a plausible range of values for the roughness evolution parameter derived from the experimental data. A framework has been developed to evaluate how the roughness evolution parameter changes over time, informed by observed changes in wind farm performance from previous studies.

**Keywords:** wind turbines; airfoil roughness; performance decline

## 1. Introduction

Wind energy has grown worldwide to become a major electricity source, which has driven the need to better understand how turbine performance varies over the 20–25 year lifetime of modern wind turbines. Understanding how performance changes as a turbines age allows for more accurate estimates of energy yield, and therefore the cost of energy, for a wind farm. It is also possible that understanding performance degradation rates, and why they occur, may present opportunities to extend turbine life through turbine design and control strategies.

Wind turbine performance declines over time because all systems are subject to deterioration as they age [1]. While there have been some advances in the diagnostics of wind turbine faults, relatively little research has been completed on the fault prognostics of wind turbines [2]. Research on individual components has indicated that the gearbox, generator, main bearing, blades, and tower are critical components for turbine reliability [3].

Despite the increasing knowledge around the failure rates of, and downtime due to, individual components, there is less understanding of performance reduction caused by degradation of the aerodynamic surfaces, or 'roughness', of the turbine blades. Roughness that occurs on rotor blades may arise due to causes such as erosion, ice accretion, and insect contamination. The effect on airfoil and turbine performance from each of these causes can vary depending on operating conditions and environmental factors, making it difficult to quantify.

A number of experimental tests have been completed detailing the effects of roughness on the performance of airfoils used for wind turbines. Janiszewska et al. [4] show the S814 airfoil has a 25% decrease in the maximum lift and a 60% increase in the minimum drag when exposed to a leading edge grit roughness pattern that was designed to emulate a wind turbine field sample. This investigation was part of a larger National Renewable Energy Laboratory (NREL) study that completed similar tests on 13 separate airfoils.

Sareen et al. [5] studied the effect of erosion on the DU 96-W-180 airfoil in wind tunnel experiments, with the most heavily eroded case resulting in a decrease in lift of 17% and an increase in drag of up to 500% at the angle of attack for the maximum lift to drag ratio of the clean (erosion free) airfoil.

The impact of ice accretion on the aerodynamic performance of an airfoil has been tested experimentally in multiple studies, showing comparable changes in lift and drag [6–8]. The performance of the NREL S826 airfoil was investigated by Hann et al. [8] and Krøgenes and Brandrud [7] under different icing conditions, showing a 10% decrease in lift and 80% increase in drag. Furthermore, Jasinski et al. [9] conducted wind tunnel testing of the S809 airfoil and used the data to simulate a 20% performance decrease on a 450 kW wind turbine, Hudecz et al. [10] tested the NACA 64-418 airfoil at a constant angle of attack ($\alpha$) and showed a decrease in lift coefficient of 22–34% for differing icing conditions, and Blasco et al. [6] tested the DU 93-W-210 airfoil showing a loss in lift of 16–25% and an increase of drag of 80–220% for different conditions.

While experimental studies, such as those reviewed above, provide valuable information about the effects of airfoil roughness, one of the key challenges is the limited number of airfoils for which there are sufficient data publicly available. Some studies have no or limited performance data [11,12] or only provide data for a single angle of attack [10,13]. This presents a challenge when trying to incorporate these data with wind turbine models, which typically require lift and drag data across a range of angles of attack. A further issue is that airfoil roughness is computationally challenging to simulate because the non-homogeneous and three-dimensional airfoil roughness geometry is difficult to define and computationally prohibitive to resolve [14,15]. Additionally, wall-modelling turbulence closures to computational fluid dynamics (CFD) models are not validated for roughness applications [15]. To overcome the difficulties in producing robust roughened airfoil data, the present study aggregates all available experimental airfoil roughness data in order to provide insights into the generalisable performance trends of roughened airfoils that can be extrapolated to other airfoils.

Experimental airfoil data is sourced from research conducted at Ohio State University (OSU) under contract from NREL [16], Sandia National Laboratories (Sandia) [17], Hann et al. [8], Blasco et al. [6], Jasinski et al. [9], and Sareen et al. [5].

Using the existing experimental data, a novel 'roughness evolution parameter' is proposed in this study that can be applied to clean (non-roughened) airfoil data to synthesise roughened airfoil data. This parameterisation can be updated as more experimental roughened airfoil data become available in future studies. The impacts of airfoil roughness are then evaluated for the DTU 10 MW reference wind turbine (RWT).

This paper is structured as follows: Section 2 describes the data sources used, Section 3 details the effect of roughness on airfoil performance and defines the roughness evolution parameter, and in Section 4 the application of the roughness evolution parameter to the DTU 10 MW RWT is demonstrated to analyse the impacts of airfoil roughness on wind turbine performance.

## 2. Roughened Airfoil Data

This section summarises the available airfoil data and their sources. All airfoil data were collected from publicly available sources, as detailed below. For all datasets, only the data from the highest chord-based Reynolds number ($Re = \rho V c / \mu$) experiment with a sufficiently wide range of angle of attack, $\alpha$ (at least $-5° \leq \alpha \leq 10°$) was used in the present study, where $\rho$ is air density, $V$ is wind speed relative to the airfoil section, $c$ is the airfoil chord length and $\mu$ is the viscosity of the air. The non-dimensional roughness height is defined relative to the chord length, $k/c$, where $k$ is the roughness height. While the magnitude of the experimental chord-based Reynolds numbers is in the range $10^5$ to $10^6$, compared to a $10^7$ for a utility-scale turbine, the presence of roughness is expected to trip the boundary layer into turbulent flow near the leading edge of the airfoil. Consequently,

the roughened airfoil experimental data are expected to provide a reasonable comparison to the conditions experienced by a utility-scale wind turbine.

### 2.1. Comparing the Effect of Roughness

Lift ($l$) and drag ($d$) forces per unit span, as functions of angle of attack, for all airfoils are converted to non-dimensional lift and drag coefficients, $c_l = l/((1/2)\rho V^2 c)$ and $c_d = d/((1/2)\rho V^2 c)$, respectively. The effects of roughness are evaluated using the percentage change in the lift and drag coefficients between clean and rough conditions, shown in Equations (1) and (2) respectively, where $\Delta$ represents the percentage change of these metrics.

$$\Delta c_l(\alpha) = \frac{c_{l,clean}(\alpha) - c_{l,rough}(\alpha)}{c_{l,clean}(\alpha)} \cdot 100\% \tag{1}$$

$$\Delta c_d(\alpha) = \frac{c_{d,rough}(\alpha) - c_{d,clean}(\alpha)}{c_{d,rough}(\alpha)} \cdot 100\% \tag{2}$$

Taking the percentage change value allows for aerodynamic differences in lift and drag magnitudes to be shown, as well as allowing comparisons with other experimental datasets that might not be produced in directly comparable experimental conditions. The mean, median, and variance of the relative lift and drag changes are then calculated.

### 2.2. Available Airfoil Datasets

Airfoil data for both aerodynamically clean and rough conditions are used in this study. Experimental data were available from six sources; data published from Ohio State University (OSU) research under contract from the National Renewable Energy Laboratory (NREL) [16], Sandia National Laboratories (Sandia) [17], Hann et al. [8], Jasinski et al. [9], Sareen et al. [5], and Blasco et al. [6]. In total, 51 datasets and 16 airfoils are used in this study, and a summary of the data used in this paper is presented in Table 1.

**Table 1.** Summary of the datasets, and their sources, used in this study. "Airfoil(s)" lists the airfoils used in the dataset, $Re$ is the Reynolds number of the data retrieved for the present work, $k/c$ indicates the roughness height(s) applied to each airfoil in a given study and "Cases" indicates the number of roughness configurations applied to each airfoil in that study. For some studies, the roughness height $k/c$ was unable to be defined due to the manner of leading edge geometry modification.

| Source | Airfoil(s) | $Re$ ($\times 10^6$) | $k/c$ | Cases |
|---|---|---|---|---|
| OSU [16] | LS417, LS421, S801 NACA4415, S809, S810 S812, S813, S814 S815, S824, S825 | 1.25 to 1.5 | $1.9 \times 10^{-3}$ | 1 |
| Sandia [17] | NACA63$_3$418, S814 | 3.2 | $1.23 \times 10^{-4}$ $1.72 \times 10^{-4}$ $2.46 \times 10^{-4}$ | 5 |
| Hann et al. [8] | S826 | 0.2 to 0.6 | N/A | 6 |
| Jasinski et al. [9] | S809 | 1.5 to 2 | $9 \times 10^{-4}$ $1.9 \times 10^{-3}$ | 5 |
| Sareen et al. [5] | DU 96-W-180 | 1.85 | N/A | 13 |
| Blasco et al. [6] | DU 93-W-210 | 1.5 | N/A | 6 |

#### 2.2.1. OSU

The OSU study has publicly available data [16] for 13 airfoils (the L303 airfoil has not been included in the results due to it being a largely cylindrical shape). A standard roughness pattern was developed from a molded insect pattern taken from a wind turbine in the field [4] and applied to all airfoils. To simulate leading edge accumulation of roughness, the density of roughness particles ranged from 5 particles per cm$^2$ in the centre

of the pattern to 1.25 particles per cm$^2$ at the edge. Based on average particle size from the field specimen, the non-dimensional roughness height was $k/c = 1.9 \times 10^{-3}$ [4].

The OSU wind tunnel experiments were conducted across a range of chord-based $Re = 0.75 \times 10^6$ to $1.5 \times 10^6$ for each airfoil.

A minor alteration to this dataset is performed in order to effectively use the data in the present study. Repeated lift/drag data points at a common $\alpha$ value are averaged. The repeated values always occurred at the point of the lowest drag.

These data have also been used for other airfoil roughness studies by Munduate and Ferrer [14], Mendez et al. [18], and Kelly et al. [15].

### 2.2.2. Sandia

Sandia has conducted studies on leading edge erosion [17] that include wind tunnel testing for two airfoils: the NACA63$_3$418 and S814. These tests were completed between $Re = 1.6 \times 10^6$ to $4.0 \times 10^6$, at three different roughness heights (100, 140, and 200 μm) and three different roughness densities (3, 9, and 15% airfoil coverage). These data are publicly available from the US Department of Energy [19]. For the purpose of this paper, the $Re = 3.2 \times 10^6$ datasets were used for both airfoils as it was the highest $Re$ data that had an adequate $\alpha$ range (at least $-5° < \alpha < 10°$).

The $k/c$ values for the three roughness heights in the Sandia study are $1.23 \times 10^{-4}$, $1.72 \times 10^{-4}$ and $2.46 \times 10^{-4}$, respectively.

### 2.2.3. Hann et al.

Hann et al. recently explored the performance penalties associated with leading-edge icing on the S826 airfoil using both experimental and numerical techniques [8]. They completed experimental tests on six geometries at $Re$ ranging between $2 \times 10^5$ to $6 \times 10^5$. Three ice geometries were produced from experimental tests at the icing wind tunnel of the Technical Research Centre of Finland (VTT) [20]. The ice geometries were then digitised by manually tracing their outlines [21]. The other three geometries were generated using a simulation tool, LEWICE [22]. The experimental ice shapes are reproduced from Hann et al. [8] in Figure 1a, and the LEWICE ice shapes in Figure 1b.

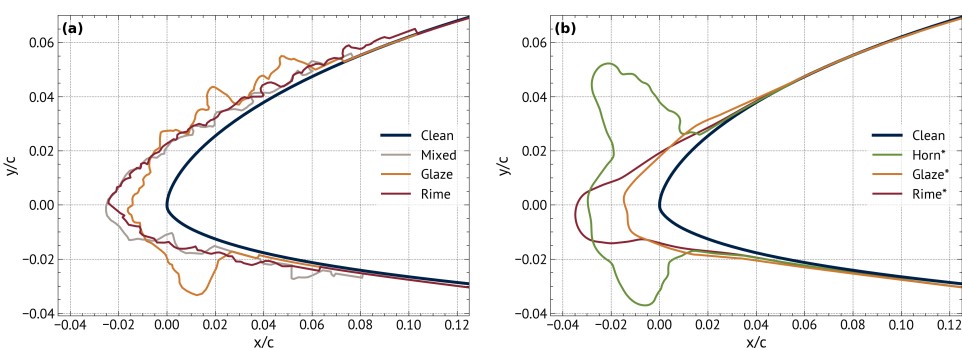

**Figure 1.** Experimental ice shapes (**a**) and LEWICE ice shapes (**b**), sourced from Hann et al. [8].

Data has been obtained for all six geometries at the highest available $Re$ ($6 \times 10^5$). Given the method used to produce the roughness seen in Figure 1, it is difficult to define a roughness height for these shapes, due to its irregularity and a change in airfoil shape.

### 2.2.4. Jasinski et al.

Jasinski et al. conducted experimental tests for four different supercooled fog conditions on the S809 airfoil [9]. The rime ice accretions were predicted using the LEWICE code, and the Reynolds number for the wind tunnel testing was in the range $Re = 1.0 \times 10^6$ to $2.0 \times 10^6$.

Data tables for lift and drag testing for the experimental tests shown in the Jasinski et al. paper (Figures 4 and 5 from the Jasinski et al. paper have been used in this study) were made

available by the corresponding author and have been used in this study. In these figures, experimental results from roughness heights of $k/c = 9 \times 10^{-4}$ and $k/c = 1.9 \times 10^{-3}$ (matching the OSU roughness height) are presented. Consistent with the other datasets, only the highest $Re$ tests were used in this study ($2 \times 10^6$ for Figure 4 and $1.5 \times 10^6$ for Figure 5 in Jasinski et al.).

### 2.2.5. Blasco et al.

As part of a study to quantify the power loss of a representative 1.5 MW wind turbine under icing conditions, Blasco et al. conducted wind tunnel testing to evaluate airfoil performance under various conditions [6]. Experimental tests were completed on the DU 93-W-210 airfoil, and six icing configurations were tested at $Re = 1.5 \times 10^6$. Icing conditions for these tests were suggested by collaborators at the National Center for Atmospheric Research, to replicate conditions experienced in the northern USA.

Roughness heights and shapes were recorded for each of the experiments, but due to the non-homogeneous and three-dimensional shapes, a $k/c$ value was not assigned for each case. The experimental data tables are available in Blasco [23].

### 2.2.6. Sareen et al.

Sareen et al. conducted wind tunnel testing to investigate the effect of leading edge erosion on the aerodynamic performance of the DU 96-W-180 airfoil [5]. Experimental tests were performed between $Re = 1.0 \times 10^6$ to $1.85 \times 10^6$ under varying erosion conditions. Bug damage was also simulated on the airfoil to assess the impact of insect accretion on airfoil performance.

Roughness height was not provided, however can be inferred from the depth of the erosion on the blade, which is given in Table 1 of [5]. These roughness heights vary from other experimental tests because the roughness applied here is subtractive rather than additive to the surface. Only data from the $Re = 1.85 \times 10^6$ experimental tests are used in this study.

### 2.3. Data Comparison

The purpose of this section is to summarise the direct comparisons that can be made across the datasets used in study. The S814 airfoil is common across both the OSU and Sandia datasets and can therefore be used for comparison. Figure 2 plots the lift and drag comparison between the two wind tunnel datasets (OSU $Re = 1.5 \times 10^6$ and Sandia $Re = 1.6 \times 10^6$), showing good agreement in lift and a systematic difference in drag in the range $-10° \leq \alpha \leq 10°$. We have adopted relative change metrics in this paper to minimise the potential impacts of systematic differences between different experimental datasets.

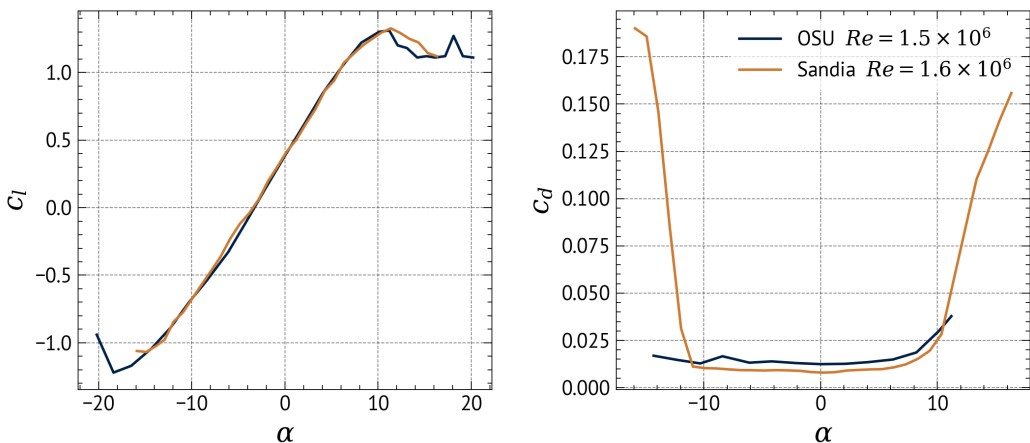

**Figure 2.** Experimental wind tunnel data comparison between OSU study at $Re = 1.5 \times 10^6$ and Sandia study at $Re = 1.6 \times 10^6$.

The S809 airfoil is common amongst this study and the OSU studies, and experimental results are validated in Figure 2 of the Jasinski et al. paper [9].

### 3. Roughness Effects on Airfoil Performance

The percentage changes in $c_l$ and $c_d$ for all datasets described in Section 2 are shown in Figure 3. The scatter points indicate the changes in lift and drag at the angle of attack $\alpha$ corresponding to the maximum lift-to-drag ratio $((c_l/c_d)^*)$ for the clean airfoil. Different symbols are used to represent the different studies included herein, and $(c_l/c_d)^*$ for all the airfoils in this study occurred at angles of attack between $3.7° \leq \alpha \leq 8.0°$. It is observed that there is a greater relative change in drag than in lift as the level of roughness develops.

One effect of roughness can be to change the angle of attack at which the maximum lift-to-drag ratio occurs from the clean case, and there can additionally be spanwise variation in angle of attack along the blades [15]. Consequently, Figure 3 also shows the changes in lift and drag in a range of $\alpha = \pm 2°$ around the angle of attack which maximises the clean lift-to-drag ratio. These ranges are shown as grey lines and intersect with the corresponding scatter point. This reinforces the observation that the relative changes in drag are greater than those for lift.

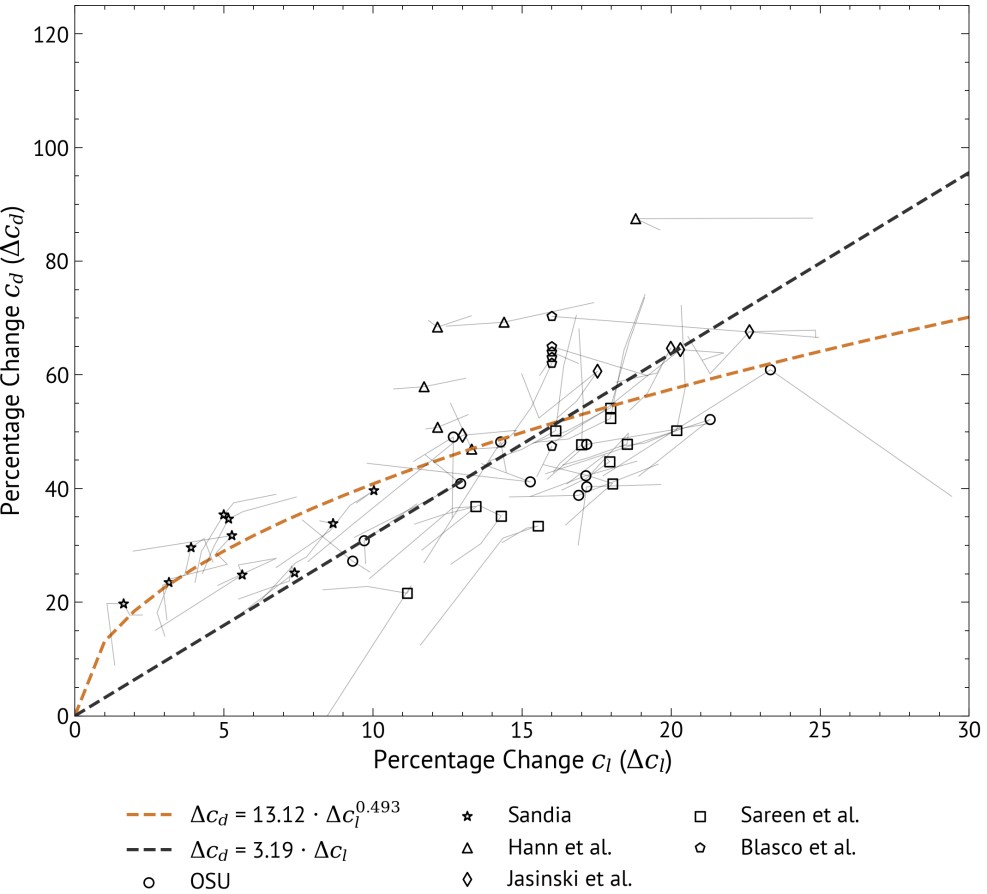

**Figure 3.** Percentage change in $c_l$ is plotted against percentage change $c_d$ for all datasets at the angle of attack for the maximum clean lift-to-drag ratio $((c_l/c_d)^*)$ point with linear (black) and power law (orange) best fit lines, plus the surrounding angles ($\pm 2°$) indicated in grey.

The general relationship between the relative change in lift and relative change in drag can be described with a linear or a power-law model, as shown in Figure 3. These models have been evaluated for the data at the optimal angle of attack of the clean airfoil (shown as scatter points in the figure), with the constraint that the fits must pass through the origin. The equation fits were chosen based on minimising the mean squared error (MSE) and the

mean absolute error (MAE). The two equations and the quality of their fit to the data are described in Table 2.

**Table 2.** Summary of the equation fits seen in Figure 3. Note, for the equations $\Delta$ represents percentage change in the corresponding metric.

| Curve Fit | Equation | Eq. Number | MSE | MAE |
|-----------|----------|------------|-----|-----|
| Power law | $\Delta c_d = 13.12 \cdot \Delta c_l^{0.493}$ | (3) | 118.42 | 8.62 |
| Linear | $\Delta c_d = 3.19 \cdot \Delta c_l$ | (4) | 166.03 | 10.91 |

Both Equations (3) and (4) demonstrate that roughness has a larger effect on the relative increase in drag than the relative decrease in lift. Roughness has a larger impact on the skin friction component of drag than the pressure component which also affects the airfoil lift, as demonstrated by the rapid increase in drag at low levels of roughness, which is better captured by the power-law fit. While there are differences for a given roughness height, the curve-fits show that as an airfoil becomes increasingly roughened, it will tend to experience a reduction in lift and increase in drag. Furthermore, the changes in lift and drag tend to be greater for airfoils with larger roughness heights and greater roughness extent. The rate at which airfoil performance declines due to roughness is not expected to be constant in time, but rather to decrease over time, that is $\partial^2(\Delta c_l)/\partial t^2 < 0$ and $\partial^2(\Delta c_d)/\partial t^2 < 0$.

While it is intuitive that changes in lift and drag due to roughness will vary as a function of time, it is difficult to quantify and is likely to depend on environmental factors. However, by using the existing roughness datasets, a general 'roughness evolution parameter' can be created which can be used to convert clean lift and drag data to roughened data. This enables the effect that roughness could have on an airfoil or wind turbine to be evaluated without any roughened data being available. An equation for modifying airfoil data using a roughness evolution parameter, $\gamma$, can then be defined based on the power law model (Equation (5)). $\gamma$ is a parametric quantity that allows change in airfoil performance to be linked to the change in roughness over time. $\gamma$ is a percentage value defined to be equal to the $\Delta c_l$ (e.g., 20% decrease in lift is represented as 20) and changes over time as a turbine becomes more roughened and operating conditions change. The available data can be used as a guide to help choose a value of $\gamma$ based on an assigned or assumed roughness height; this is discussed further in Section 4.3.

$$c_{d,rough} = \left(1 + \frac{13.12 \cdot \gamma^{0.493}}{100}\right) \cdot c_{d,clean}$$

$$c_{l,rough} = \left(1 - \frac{\gamma}{100}\right) \cdot c_{l,clean} \tag{5}$$

Figure 4 shows the percentage change in lift and drag for different roughness heights aggregated across the datasets where $k/c$ was defined. This shows that as roughness height progressively increases, lift decreases (i.e., $\Delta c_l$ increases), and the data moves up the best fit line as the drag increases. The development of airfoil roughness over time corresponds to moving towards the right of the figure.

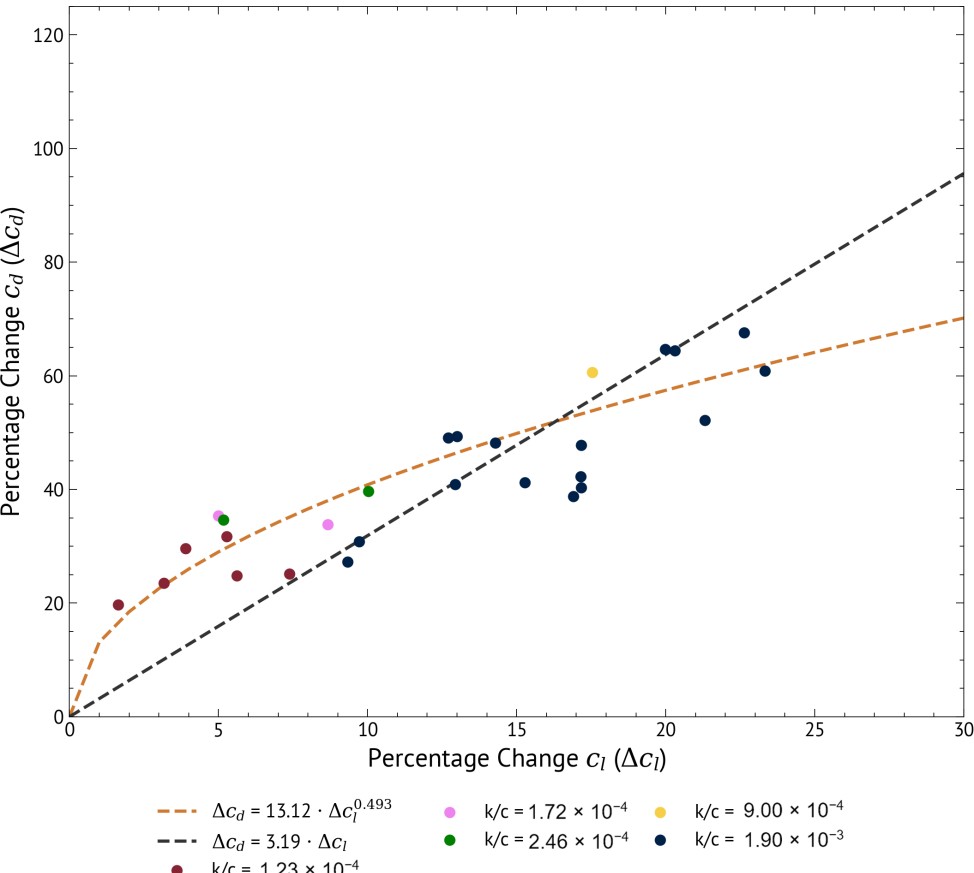

**Figure 4.** Percentage change in lift is plotted against percentage change in drag for the OSU, Sandia and Jasinski et al. datasets with the linear (black) and power law (orange) best fit lines superimposed. The legend indicates the roughness height.

## 4. Roughness Impacts on Wind Turbine Performance

This section details the methods and outcomes of combining and applying the roughened airfoil data results to the DTU 10 MW reference wind turbine (RWT) to investigate the impacts of roughness on turbine performance. The (clean) airfoil data for the RWT is modified according to the model developed in Section 3 to create roughened airfoil data, which is then applied to the full rotor using Blade Element Momentum (BEM) theory.

### 4.1. Turbine Model

The BEM model used in this study is based on the single parameter formulation with guaranteed convergence proposed by Ning [24]. The model has been validated with reference to BEM-based studies, see Kelly et al. [15] and Vogel et al. [25] for further details. Prandtl's tip-loss model [26] is included to account for the influence of discrete blade effects that are not directly modelled in the 2-D momentum and blade element equations used in BEM theory. Buhl's thrust correction [27] for the turbulent wake state that couples momentum theory with Glauert's empirical model for heavily loaded turbines with tip losses is also included.

The DTU 10 MW RWT was designed to provide a realistic, publicly available reference rotor for academic applications [28]. The DTU 10 MW RWT has a hub height of 119 m, rotor diameter of 178.3 m, a cut-in wind speed of $U_{ci} = 4$ m/s, a rated wind speed of 11.4 m/s, and a cut-out wind speed of $U_{co} = 25$ m/s. In this study the DTU 10 MW RWT is modelled using the aforementioned in-house BEM method, with the publicly available airfoil and rotor data from DTU.

### 4.2. Performance Change Metrics

Performance loss for the DTU 10 MW RWT is calculated by comparing turbine performance ($C_P$, $C_T$, and AEP) with clean and roughened airfoils. Note that the roughened airfoils are applied to the entire span of the rotor, implying a uniform distribution of roughness along the blade. While this is likely a simplified representation of blade roughness, it will indicate the general effects of roughness on turbine performance.

$C_P$ was calculated by assessing turbine performance relative to the undisturbed kinetic energy flux, using Equation (6):

$$C_P = \frac{P}{\frac{1}{2}\rho U^3 \pi (R^2 - R_h^2)} \tag{6}$$

where $P$ is turbine power, $R$ is blade radius, and $R_h$ is hub radius. Similarly, $C_T$ was calculated by comparing thrust against the undisturbed momentum flux, using Equation (7):

$$C_T = \frac{T}{\frac{1}{2}\rho U^2 \pi (R^2 - R_h^2)} \tag{7}$$

where $T$ is turbine thrust. AEP was assessed through numerical integration of Equation (8), where $\tau$ is the time period (one year), $P(U)$ represents the power as a function of wind speed, $f(U)$ represents the likelihood of a wind speed $U$ occurring at a particular site, integrated across the range of operating wind speeds.

$$E = \tau \int_{U_{ci}}^{U_{co}} P(U) f(U) dU \tag{8}$$

The wind speed distribution $f(U)$ is evaluated using a two-parameter Weibull distribution [29]:

$$f(U; k_s, c_s) = \frac{k_s}{c_s} \left( \left( \frac{U}{c_s} \right)^{k_s - 1} \right) \exp \left( - \left( \frac{U}{c_s} \right)^{k_s} \right) \tag{9}$$

where $k_s$ is the unitless shape parameter and $c_s$ is the scale parameter in m/s.

Wind speed data from the Kentish Flats wind farm site in the UK was used in this study, and the data were retrieved from the Crown Estate's marine data exchange [30]. Kentish Flats is an offshore site and the wind data are from the 2002–2005 period, before the wind farm was built. The data were recorded at an altitude of 80 m, in 10 minute average wind speeds, and have been extrapolated to hub height assuming a neutrally-stable log law profile. This data was best fit to a Weibull distribution with a shape parameter, $k = 2.83$, and a scale factor, $c = 10.52$; 68% of the distribution lies between $U_{ci} \leq U < U_{rated}$, 19% lies between $U_{rated} \leq U < U_{co}$, with the remaining 13% of the time lying between $U < U_{ci}$ or $U \geq U_{co}$ where the turbine is shut down. A different wind speed distribution, e.g., for a different location, will affect AEP by changing how often the turbine is in pre-rated, post-rated, or non-operational configurations; however, the underlying aerodynamic mechanisms causing the performance drop will not be affected.

Effect of Roughness on DTU 10 MW RWT

Figure 5 displays the change in lift against change in drag values from all datasets as well as the modelled (a) $C_P$, (b) percentage change in AEP and (c) $C_T$ values for the DTU rotor for the corresponding change in lift and drag. Also included are the lines of best fit from Figure 3, where moving towards the right along the curves indicates an increasing level of roughness.

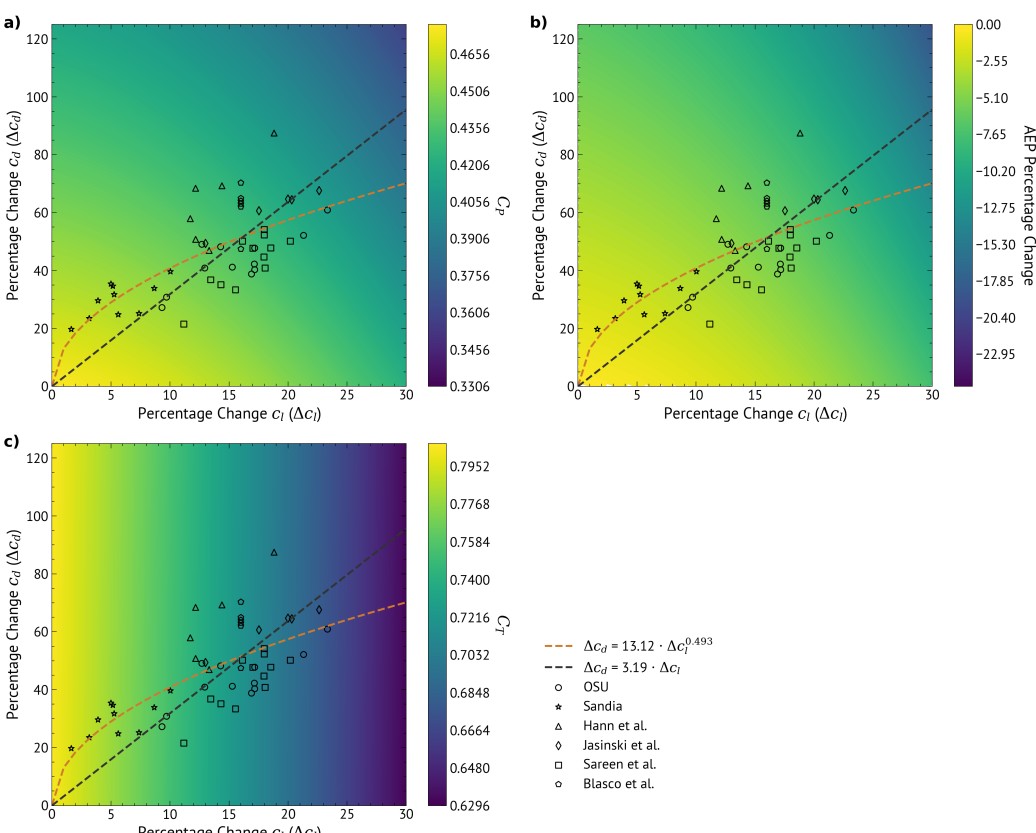

**Figure 5.** Percentage change in lift is plotted against percentage change in drag for all datasets with linear (black) and power law (orange) best fit lines are shown in all plots, and contour plots of the modelled $C_P$ values (**a**), percentage change AEP (**b**), and $C_T$ values (**c**) for the DTU 10 MW RWT.

Taking values from both ends of the experimental range, a plausible range of $\gamma$ can be assumed to be $1 < \gamma < 25$ (i.e., a $\Delta c_l$ of 1.00–25.00% and a $\Delta c_d$ of 13.12–64.14% using the power law fit). Using this range, we can investigate different roughness cases for the DTU 10 MW RWT to evaluate the effect on performance. Two separate cases, $\gamma = 1.00$ and $\gamma = 25.00$, are applied to the clean airfoil lift and drag data in the range $1° \leq \alpha < \alpha_{sep}$, where $\alpha_{sep}$ is the angle where separation starts to occur in clean data. This results in a $C_P = 0.472$ ($\Delta C_P = -0.005$), $C_T = 0.799$ ($\Delta C_T = -0.005$), and an AEP reduction of 0.6% for the $\gamma = 1$ case, and a $C_P = 0.407$ ($\Delta C_P = -0.070$), $C_T = 0.664$ ($\Delta C_T = -0.140$), and a percentage AEP reduction of 9.6%, for the $\gamma = 25$ case, respectively. This gives a plausible range of performance decrease when the DTU 10 MW rotor blades are slightly ($\gamma = 1.00$) and severely ($\gamma = 25.00$) roughened.

The reduction in $C_P$ as roughness increases means that the energy capture efficiency at lower wind speeds is reduced, and that the turbine reaches rated power at progressively higher, and less frequent, wind speeds. Consequently, the AEP worsens non-linearly as the level of roughness on the blades increases. The interaction between blade roughness and turbine control is discussed in more detail in, for example, [15].

### 4.3. Impact of Roughness on Performance over Time

The above analysis has yielded a roughness evolution parameter that can be applied to clean airfoil data to synthesize the effects of roughness where prior experimental data does not exist. To both contextualise the roughness evolution parameter values and relate it to energy yield, the AEP performance decline shown in Figure 5b can be related to observed wind farm performance decline.

Multiple studies have explored how wind farm energy yield declines as they age. Staffell and Green [31] show that UK wind farms decline on average at 1.6% per year,

Olauson et al. [32] show a 0.63% annual performance decline for Swedish wind farms, and Germer and Kleidon [33] show a 0.6% annual performance decline for German wind farms. These decline rates over time are shown in Figure 6a. These performance decline curves can be compared with the impact that roughness has on turbine AEP to estimate the AEP decline over time. It is assumed that the impact of wind farm wake effects does not change over time. This analysis then assumes for simplicity that the decline in performance is completely attributable to blade roughness, and that the rate at which outages (e.g., due to maintenance or curtailment) occur does not change over time. The implications of these assumptions are discussed further in Section 5.

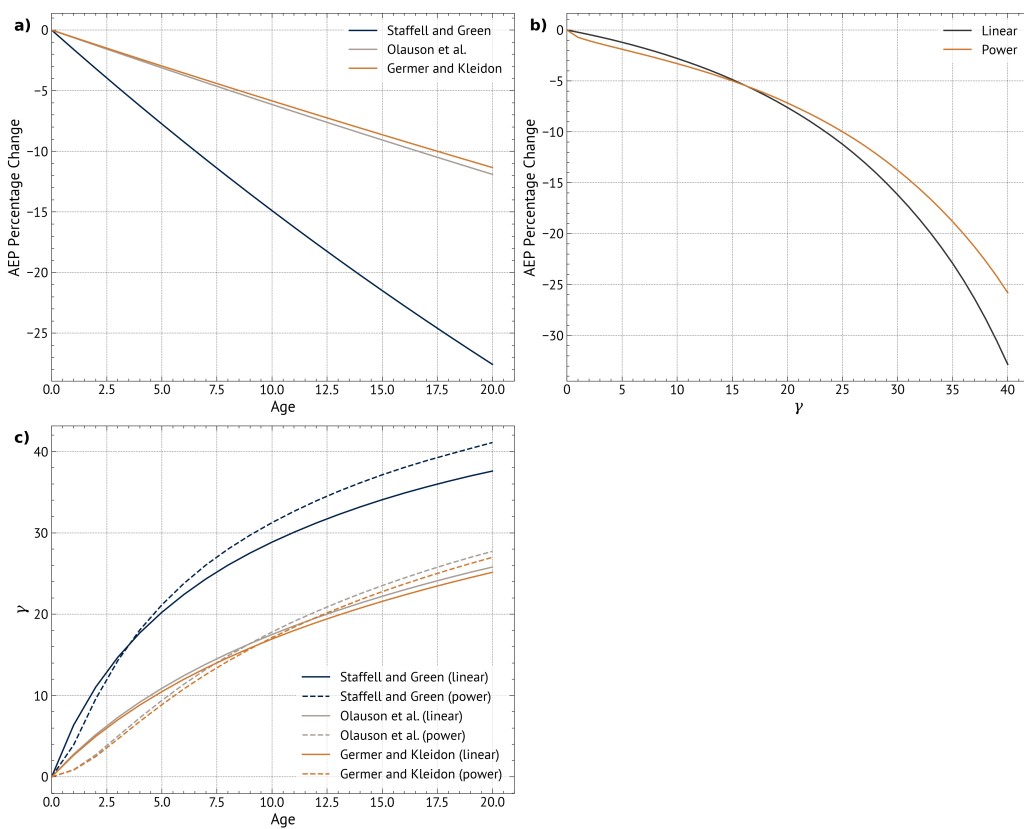

**Figure 6.** Wind farm age against AEP curves from Staffell and Green [31], Olauson et al. [32], and Germer and Kleidon [33] (**a**). $\gamma$ against AEP as calculated using BEM analysis of the DTU 10 MW RWT (**b**). Wind turbine age against $\gamma$ (**c**).

The roughness evolution parameter $\gamma$ can be plotted against AEP percentage change, using the intersection between the contours and best fit relationships for the DTU 10 MW RWT from Figure 5b to produce Figure 6b. Under the assumption that changes in AEP over time are due to the development of roughness on turbine blades, a relationship between the age of a wind farm and $\gamma$ is found using the common y-axis of Figure 6a,b. This demonstrates that the increase in the roughness evolution parameter gradually slows over time as the roughness becomes more developed on the blade. The rate of change is likely to be dependent on location, with offshore turbines more affected by the accretion of ice on the blade, whereas onshore turbines will tend to be more affected by insects and erosion due to atmospheric dust. Combining Figure 6b and Figure 6c can be used to relate wind turbine AEP performance loss due to roughness to turbine age. This relationship is shown in Figure 7, which bears a strong similarity to Figure 6a—wind farm AEP variation over time—due to the assumption that the time variation in turbine performance loss is due entirely to airfoil roughness. This will vary with modified assumptions about how other factors affecting turbine performance that are beyond the scope of the present study, such as component failure, vary with age.

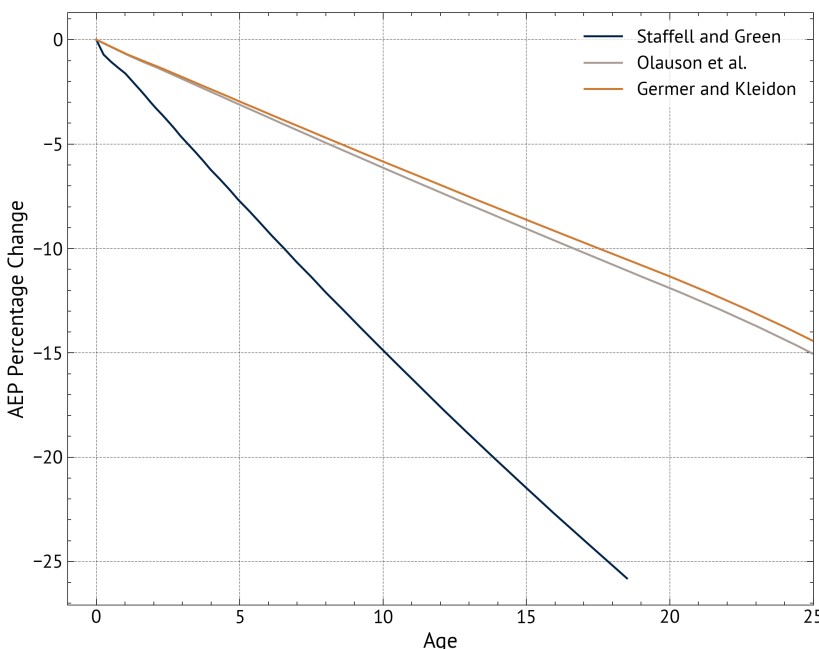

**Figure 7.** Wind turbine age against AEP performance change due to roughness based on wind farm performance decline rates from Staffell and Green [31], Olauson et al. [32], and Germer and Kleidon [33].

## 5. Discussion and Conclusions

A method has been created through the use of existing experimental data to create a 'roughness evolution parameter'. This parameter can be applied to clean airfoil data to create the corresponding roughened data, which can then be applied to a rotor to examine the effect of blade roughness on turbine performance. This is useful because there is currently a lack of experimental data on roughened airfoils, which limits turbine studies using new airfoil geometries. A demonstration case was investigated using the DTU 10 MW RWT that resulted in an AEP reduction range of 0.6–9.6%.

The creation of a roughness evolution parameter is a novel finding but would benefit from further data to verify the relationship. More experimental tests on airfoils with roughness applied should occur to evaluate the change in $c_l$ and $c_d$. As this study has demonstrated, this should be completed for both a range of airfoil shapes and at different roughness heights. This is important to provide additional information on the relationship between change in lift and change in drag and allow further refinement of the roughness evolution parameter.

A framework for converting the roughness evolution parameter to AEP reduction over time has been proposed. The framework relates predictions of AEP (accounting for roughness) to observed changes in AEP from wind farms. Building on the wind farm performance decline work of Staffell and Green [31], Olauson et al. [32], and Germer and Kleidon [33], the turbine AEP percentage change due to roughness was estimated to be in the range of 2.5–7.5% at age 5 and 8.5–22.5% at age 15, depending on which previous study is used.

This study has attributed the time-varying component of performance decline in wind farms to an increase in turbine blade roughness over time, which implies that the impact of factors such as wind farm wake interactions and component failure is constant in time. This simplifying assumption neglects the potential change in wake interactions as turbine performance changes and that probability of component failure is likely to increase non-linearly with turbine age.

Despite these simplifications, this study aims to provide a framework for a method used to relate roughness to performance decline. Alternative assumptions about how wind

turbine performance varies with age can be incorporated through modifications to the curves presented in Figure 6.

The framework for converting the roughness evolution parameter to AEP reduction over time gives important context to the impact of roughness on performance. By providing a framework method, it allows future researchers to refine the assumptions about the impacts of roughness on AEP reduction as additional data become available.

**Author Contributions:** Conceptualization, J.K., C.V. and R.W.; methodology, J.K.; validation, C.V. and R.W.; formal analysis, J.K.; resources, J.K.; data curation, J.K.; writing—original draft preparation, J.K.; writing—review and editing, J.K. and C.V.; project administration, J.K. All authors have read and agreed to the published version of the manuscript.

**Funding:** The authors would like to acknowledge The Australia Day Foundation UK Trust for their support for JK in conducting this study, and EPSRC who support RHJW's research through an EPSRC Advanced Fellowship EP/R007322/1 and the EPSRC Supergen ORE Hub, grant number EP/S000747/1. CRV acknowledges the support of the UKRI through his Future Leaders Fellowship MR/V02504X/1.

**Institutional Review Board Statement:** Not applicable.

**Informed Consent Statement:** Not applicable.

**Data Availability Statement:** The data presented in this study are openly available in Ohio State University Wind Tunnel tests at https://www.nrel.gov/wind/nwtc/airfoils-osu-data.html (accessed on 1 October 2019), Leading Edge Erosion (Sandia) at https://a2e.energy.gov/projects/lees (accessed on 1 May 2020), Hann et al. [8], Jasinski et al. [9], Blasco et al. [23], and Sareen et al. [5].

**Acknowledgments:** The authors wish to acknowledge Michael Selig and Sven Schmitz for providing tabulated airfoil roughness data from their published work.

**Conflicts of Interest:** The authors declare no conflict of interest. The funders had no role in the design of the study; in the collection, analyses, or interpretation of data; in the writing of the manuscript, or in the decision to publish the results.

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
