# Peer review of "Parameterising the Impact of Roughness Evolution on Wind Turbine Performance"

_2674-032X, doi:10.3390/wind2020022_

Round 1
Reviewer 1 Report
The linear regression model obtained by the authors (see Table 2) does not fulfil the requirements of the estimation measures used by the authors. Other methods of verification of the estimated model in terms of goodness of fit to empirical data should be applied or a different regression model should be used. R2 coefficient cannot be applied to a non-linear regression model.
Equations (3) and (4) have not been correctly estimated in terms of their fit. This is a big mistake. Correct regression models should be determined and their fit should be estimated using appropriate coefficients.
As further analyzes presented in the article are based on equations (3) and (4), they cannot be verified in terms of correctness and reliability.
The roughness evolution parameter was defined as the percentage change of the bearing capacity coefficient Delta_Cl. The authors of this parameter cannot be verified for correctness.
Other errors and comments are posted in the form of comments in the article

Author Response
Dear Reviewer,
Thank you for your comments and suggestions - please find attached our responses.
Best wishes
Jack Kelly
Christopher Vogel
Richard Willden

Reviewer 2 Report
This is a nicely written manuscript where the authors present a parametric study on the impact of roughness evolution on a wind turbine performance. The proposed work fits within the aim of the journal and constitutes a topic of high relevance. The quality of the present work is outstanding and I would like to congratulate the authors for this interesting piece of work, which I recommend to accept after considering the following minor corrections:
1. Acronyms should be avoided in the abstract.
2. The introduction should be completed adding a final paragraph outlining the content of the rest of the manuscript, so prospective readers can easily navigate through the text.
3. An additional section with symbols, acronyms and nomenclature is needed for an appropriate read of the text.
4. I would suggest to use a different Greek letter to represent the “roughness evolution parameter” as it may lead to possible misunderstanding to prospective readers and confuse it with the turbine’s performance.
The quality of the English and the fluency of the text is outstanding and I do not have further comments.
Author Response

(The authors gave the same response as above.)

Round 2
Reviewer 1 Report
The ideal value of the mean absolute error (MAE) is 0.0. The authors obtained results that deviate from this value, however, they could be considered correct. However, also the ideal mean square error (MSE) value is 0.0. In this case, the results obtained by the researchers significantly deviate from this value. It is difficult to recognize the performed statistical analysis as correct.
The lack of interpretation of the obtained values makes it impossible to understand the value of the obtained results. According to the reviewer, correct interpolation of the results should be carried out. The rest of the article is based on the equations ((3) and (4)) determined in this way.
I still believe that the main flaw of the research presented in the article has not been corrected.

Author Response
Please see our responses in the attached document.

Round 3
Reviewer 1 Report
I would like to thank the authors for their exhaustive reference to my comments. Everything is understandable now. I believe that adding a reply to my first remark to the article will significantly affect its substantive quality.
